# Background Blurring Matters: Improving Visual Grounding by Merging Text-Irrelevant Tokens

## Abstract

Visual grounding (VG) aims to precisely localize the object in input images based on its natural language descriptions. Most recently proposed methods deal with this task with transformer-based architectures that can inject the textual information into the visual features. However, due to the image tokenlization procedure, there will be a large amount of image tokens located in text-irrelevant background areas. These tokens can introduce noise into the attention calculation, thus reducing the significance of foreground object tokens and ultimately affecting the effectiveness of these methods. To this end, we propose a novel **To**ken **B**lurring (ToB) module, which dynamically merges image tokens based on the pair-wise visual similarity between them and their textual relevance with input expressions. By reducing the number of text-irrelevant background tokens and preserving the density of text-referred ones, ToB can improve both model effectiveness and efficiency in solving VG tasks. Extensive experiments on RefCOCO, RefCOCO+, and RefCOCOg datasets show that transformer-based models equipped with our ToB module yield better results while reducing computational overhead compared to various existing VG methods.

## 1 Introduction

Visual Grounding (VG) (Xiao et al., 2024a) aims to localize a target object in an input image which is described by a given textual query. It has become a fundamental task in many multi-modal reasoning applications, such as visual question answering, automatic driving and human-machine dialogue.

Most of existing VG methods (Carion et al., 2020; Deng et al., 2021; Shi et al., 2023) are typically constructed based on the Transformer architecture (Vaswani et al., 2017), which excels in modeling visual-linguistic cross-modal interactions. In general, these methods consider the VG task as a bounding box regression problem and tackle it through a dual-branch framework: the visual and linguistic features are first encoded by two separate transformer encoder branches and then fed into a shared transformer decoder, in which they are fused with each other to predict the target bounding box. However, this parallel encoding strategy prevents the visual feature extraction process from interacting with the textual information of input expressions. As a result, the visual encoder cannot distinguish the text-referred foreground regions from text-irrelevant background areas, thus limiting its representative ability when processing VG tasks.

To deal with this problem, a series of recently proposed VG approaches (Ye et al., 2022; Yang et al., 2022a;b; Su et al., 2023; Luo et al., 2024; Yao et al., 2024) have begun to explore the language-guided visual encoder structures. They attempt to inject the textual information into visual features through cross-attention mechanisms, language-derived feature modulation parameters, multi-modal adaptation weights, and so on. These operations allow the visual encoder to dynamically adjust the attention scores of its intermediate layers, so as to implicitly highlight the text-referred object regions and suppress those irrelevant to the input expressions. However, a major problem with these methods is that they only seek more effective representations for image tokens while ignoring the influence of their numbers. Specifically, the image tokens of these methods are obtained by evenly dividing the whole image into non-overlapping patches of equal size. In this way, there will be a large amount of tokens located in text-irrelevant background areas, especially when the target object

has a relatively small size. Although suppressed by the text injection operations mentioned above, these background tokens will still produce small attention scores via the softmax function, which may introduce some noise into the overall attention calculation, thus reducing the significance of foreground object tokens and ultimately affecting the effectiveness of current VG methods. In addition, simultaneously processing all the image tokens often leads to a large computational overhead, due to the quadratic complexity of transformer-based architectures.

In this paper, we propose a novel **To**ken **B**lurring (ToB) module to alleviate the above problems in solving VG tasks. Inspired by human vision that can blur the surrounding background to focus more on the target object of interest, *our ToB module also attempts to "blur" the background regions of input images by gradually merging the text-irrelevant tokens, while maximally preserving the density of text-referred foreground tokens, as illustrated in Figure 1.* To achieve this, ToB employs a language-guided token merging strategy. Specifically, it first computes a text-aware weight for each image token based on its visual-linguistic correlations with the

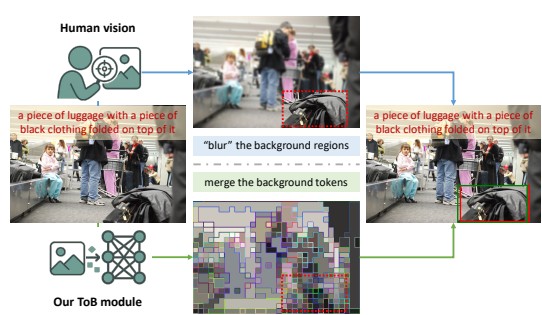

Figure 1: Inspired by human vision, our ToB module highlights foreground targets by blurring (merging) visually similar and text-irrelevant tokens in the background to achieve better detection performance.

input textual query. These weights are then combined with pair-wise visual similarities to identify $r$ token pairs that have the most similar visual features and the lowest textual relevance. Finally, each pair of tokens is merged into a new "blurred" token, their text-aware weights and visual features are mixed accordingly and then cooperate with each other to generate the embedding of this merged token. Unlike traditional token merging/selection methods (Liu et al., 2024a; Fu et al., 2024; Ye et al., 2025) that rely solely on visual similarity, ToB incorporates the textual relevance as additional guidance information, which leads to a more accurate token merging process for handling multi-modal VG tasks. In this way, as shown in Figure 1, ToB explicitly reduces the number of background tokens with lower textual relevance, thus mitigating their potential negative effects on the calculation of attention scores. In contrast, the visual details of the text-referred target object are maintained by ToB, which can help to enhance the final detection accuracy. Furthermore, ToB also allows the transformer block to process fewer image tokens, thus decreasing its computational overhead and improving the model's efficiency. We summarize the main contributions of this work as follows:

- We propose a novel module called **To**ken **B**lurring (ToB), which dynamically merges image tokens based on the pair-wise visual similarity between them and their textual relevance with input expressions. By reducing the number of text-irrelevant background tokens and preserving the density of text-referred ones, ToB can improve both model effectiveness and efficiency in solving VG tasks.
- We construct a transformer-based model based on the proposed ToB module. By adopting DINOv2-B/BERT-B as the visual/linguistic encoder, this model achieves competitive results with recently proposed benchmarks.
- We conduct extensive experiments on three widely-used VG datasets, RefCOCO, RefCOCO+ and RefCOCOg, which validate the effectiveness of ToB and the language-guided token merging strategy. Moreover, the experimental results also show that ToB can be easily integrated into other transformer-based VG models, leading to better performance.

## 2 RELATED WORK

**Visual Grounding**  Inheriting the general object detection framework, existing VG methods can be divided into two-stage methods (Yang et al., 2019a; Yu et al., 2018; Zhang et al., 2018; Liu et al., 2019) and one-stage methods (Chen et al., 2018; Yang et al., 2019b; Liao et al., 2020). Two-stage methods match the language feature to the vision content at the region level, thus requiring the vision encoder to first generate a set of region proposals. One-stage methods densely perform

multi-modal feature fusion at all spatial locations, waiving the requirements of region proposals, and predict the location of referred objects directly. With the success of Transformer in detection and vision-language tasks, a series of transformer-based methods have been proposed. TransVG (Deng et al., 2021) incorporates DETR encoder (Carion et al., 2020) to extract visual features and proposes a multi-modal reasoning module. MMCA (Yao et al., 2024) adaptively adjusts the visual encoder's weights based on multi-modal features to enhance visual representation extraction. SegVG (Kang et al., 2024) transfers the bounding box annotation into additional segmentation signals to exploit the pixel-level details of the target regions. SimVG (Dai et al., 2024) decouples multi-modal fusion from grounding tasks using pre-trained multimodal backbones, object tokens, and weight-balanced distillation to achieve both speed and accuracy. OneRef (Xiao et al., 2024c) presents a unified modality-shared, single-tower transformer architecture employing Mask Referring Modeling (MRefM) to dynamically mask irrelevant regions, thus facilitating minimalist yet effective grounding and segmentation. AttBalance (Kang et al., 2025a) dynamically imposes and balances constraints of the attention to optimize the behavior of visual features within language-relevant regions. TCRT (Chen et al., 2025) integrates large language model priors and cross-modal feature refinement to guide attention more precisely and suppress non-target background information. Recent ExpVG (Kang et al., 2025b) systematically explores the paradigm and data designs for VG in multi-modal large language models (e.g., LLaVA series). Notably, while recent improvements primarily result from language-guided attention modules focusing on highlighting foreground regions, these methods often overlook the equally crucial aspect of suppressing background interference by reducing text-irrelevant tokens explicitly.

**Token Merging**   Token merging can be broadly divided into two categories: diversity-based and task-relevant strategy. Diversity-based approaches aim to reduce visual redundancy by pruning or merging similar tokens based on visual similarity. Token pruning techniques such as EViT and Adaptive Token Sampling remove low-importance tokens using attention scores or gating mechanisms (Fayyaz et al., 2022; Liang et al., 2022; Wei et al., 2023). In contrast, token merging methods like ToMe and Token Pooling combine similar tokens through bipartite matching or clustering (Bolya & Hoffman, 2023; Marin et al., 2023; Bolya et al., 2023). Recent methods such as ToE (Huang et al., 2024) and ToFu (Kim et al., 2024) further refine these ideas by dynamically adjusting merging strategies or jointly applying pruning and merging. However, these methods primarily focus on visual similarity and overlook task-specific semantics. This makes them suboptimal for multimodal tasks such as visual grounding, where maintaining fine-grained visual-textual alignment is crucial. This highlights the need for more adaptive, context-aware token reduction strategies. On the other hand, a growing line of work introduces task-aware token selection. LAPS (Fu et al., 2024) uses linguistic supervision to retain text-relevant tokens while merging the rest, but at the cost of discarding visually rich details essential for precise localization. MustDrop (Liu et al., 2024a) applies a staged process of merging and pruning, guided by textual relevance, yet decouples visual-textual interaction across stages. FitPrune (Ye et al., 2025) minimizes representation drift during pruning but still relies on attention scores and lacks direct language integration in the merging process. Different from these previous works, our method introduces a language-guided token merging strategy that simultaneously incorporates both textual relevance and visual similarity into the merging process, dynamically merging background visual tokens according to a joint visual-textual similarity metric, enabling more precise localization in visual grounding tasks.

## 3 METHODOLOGY

### 3.1 PRELIMINARIES

Given an image and a related textual expression, the main goal of the VG task is to accurately detect the text-referred target object in this image. To this end, existing VG models are mostly established following an "encoder-decoder" paradigm. Specifically, the input image and expression are first tokenized and then separately passed through a visual encoder and a linguistic encoder to obtain their corresponding features $F_v \in \mathbb{R}^{N_v \times d_v}$ and $F_l \in \mathbb{R}^{N_l \times d_l}$. These two encoders are often implemented using the transformer architecture. $N_v$ and $N_l$ represent the number of image and language tokens, respectively, while $d_v$ and $d_l$ denote their feature dimensions. After that, $F_v$ and $F_l$ are fed into a multi-modal decoder, in which a regression token [REG] is learned to aggregate the target object related information from $F_v$ and $F_l$. Finally, a regression head takes the resulting [REG] token as input to predict the bounding box coordinates of the text-referred object.

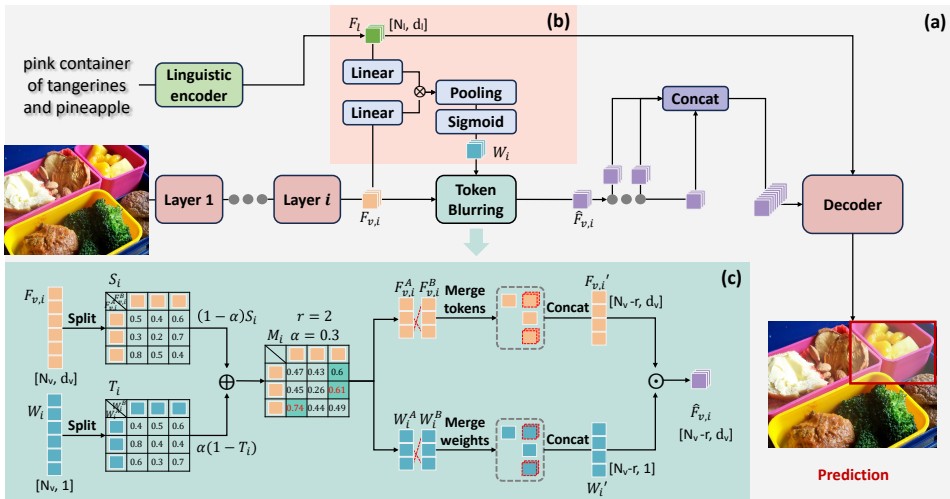

Figure 2: Overview of our proposed framework. The figure illustrates (a) the overall framework of our method, which includes (b) the process for generating text-aware weight and (c) the proposed Token-Blurring strategy. In (c), we demonstrate the Token-Blurring process using an example with $r = 2$ and $\alpha = 0.3$.

## 3.2 THE TOKEN BLURRING MODULE

Let $F_{v,i} \in \mathbb{R}^{N_{v,i} \times d_v}$ indicate the output features of the $i$-th transformer layer in the visual encoder. Our ToB module aims to merge multiple pairs of tokens that are visually similar but irrelevant to the input expression, so as to explicitly reduce the number of background tokens in $F_{v,i}$ and thereby suppress their negative impact on target object detection. To achieve this, as illustrated in Figure 2, ToB first uniformly divides $F_{v,i}$ into two non-overlapping subsets $F_{v,i}^A, F_{v,i}^B \in \mathbb{R}^{(N_{v,i}/2) \times d_v}$, corresponding to tokens at odd and even indexes, respectively:

$$F_{v,i}^A = \{f_{v,i}^1, f_{v,i}^3, \cdots, f_{v,i}^{(N_{v,i}-1)}\}, \quad F_{v,i}^B = \{f_{v,i}^2, f_{v,i}^4, \cdots, f_{v,i}^{N_{v,i}}\}, \tag{1}$$

where $f_{v,i}^j$ denotes the $j$-th token embedding in $F_{v,i}$. By performing a one-to-one matching between these two subsets, we can obtain a total of $(N_{v,i}/2) \times (N_{v,i}/2)$ token pairs, from which ToB identifies those pairs located in background areas according to a criteria of **high visual similarity** and **low textual relevance**: (1) Two image tokens with high visual similarity usually imply that they are belonging to the same or similar objects, and thus most of the information contained in their features is repetitive and redundant. Hence, merging them into a single "blurred" token will not result in a significant loss of critical information for image representation. (2) The tokens have low textual relevance typically indicating that they are semantically less relevant to the target object and most likely located in the background regions. Therefore, the tokens selected by this visual-textual joint criterion can be effectively distinguished from those object-related foreground ones.

Specifically, ToB calculates the visual similarity between the tokens in $F_{v,i}^A$ and $F_{v,i}^B$ as follows:

$$S_i = \mathcal{F}_{\text{Cosine}}(F_{v,i}^A, F_{v,i}^B), \tag{2}$$

where $\mathcal{F}_{\text{Cosine}}(\cdot, \cdot)$ denotes the pair-wise cosine similarity function.

As for the textual relevance, ToB first generates a text-aware weight for each token in $F_{v,i}$ as follows (see Figure 2(b)):

$$W_i = \mathcal{F}_{\text{Sigmoid}}(\mathcal{F}_{\text{AvgPool}}(F_{v,i} P_{v,i} (F_l P_{l,i})^{\mathsf{T}}))). \tag{3}$$

Here, $\mathcal{F}_{\text{Sigmoid}}(\cdot)$ is the sigmoid function and $\mathcal{F}_{\text{AvgPool}}(\cdot)$ represents the average-pooling operation along the text length dimension. $P_{v,i} \in \mathbb{R}^{d_v \times d_c}$ and $P_{l,i} \in \mathbb{R}^{d_l \times d_c}$ are two linear projectors for the $i$-th transformer layer that are learned to map $F_{v,i}$ and $F_l$ into a $d_c$-dimensional common space, respectively, so that their feature dimensions can be aligned. As a result, each element in $W_i \in \mathbb{R}^{N_{v,i} \times 1}$ indicates the average correlation strength between the corresponding image token and all

language tokens. As shown in Figure 2(c), by adopting the same splitting strategy in Eq. (1), $W_i$ can also be divided into two subsets $W_i^A, W_i^B \in \mathbb{R}^{(N_{v,i}/2) \times 1}$. They are used to calculate the textual relevance of a token pairs between $F_{v,i}^A$ and $F_{v,i}^B$ through the following geometric mean operation:

$$T_i = \sqrt{W_i^A (W_i^B)^{\mathrm{T}}}. \tag{4}$$

Then, by introducing a positive parameter $\alpha$ to balance the relative importance of $S_i$ and $T_i$, we linearly combine them and obtains a score matrix $M_i \in \mathbb{R}^{(N_{v,i}/2) \times (N_{v,i}/2)}$ for identifying background token pairs as follows:

$$M_i = (1 - \alpha)S_i + \alpha(1 - T_i). \tag{5}$$

With this definition, an element in $M_i$ with a higher value often means that its related two token have a higher visual similarity and are more likely both located in text-irrelevant background areas, so they should have a higher priority to be merged in the subsequent steps. Based on this score matrix, ToB performs a bipartite matching operation between $F_{v,i}^A$ and $F_{v,i}^B$. Specifically, for each token $f_{v,i}^{A,m}$ in $F_{v,i}^A$, we search for the highest score in its corresponding row of $M_i$ so as to find its paired token $f_{v,i}^{B,n}$ in $F_{v,i}^B$, resulting in $N_{v,i}/2$ candidate token pairs.

After that, ToB ranks these token pairs in descending order according to their scores, and selects the top $r$ ones for the following token merging operations:

$$\mathcal{F}_{\mathrm{Merge}}(f_{v,i}^{B,n}) = \frac{f_{v,i}^{B,n} + \sum_{m \in E_i^n} f_{v,i}^{A,m}}{|E_i^n| + 1}, \tag{6}$$

$$\mathcal{F}_{\mathrm{Merge}}(w_i^{B,n}) = \frac{w_i^{B,n} + \sum_{m \in E_i^n} w_i^{A,m}}{|E_i^n| + 1}. \tag{7}$$

Here, $E_i^n$ refers to a set recoding the indexes of tokens $f_{v,i}^{A,m}$ in the selected $r$ token pairs from $F_{v,i}^A$ that are associated with $f_{v,i}^{B,n}$ from $F_{v,i}^B$. $|E_i^n|$ denotes the total number of elements in $E_i^n$. In Eq. (6), all the tokens $f_{v,i}^{A,m}$ indexed by $E_i^n$ are merged with their paired token $f_{v,i}^{B,n}$ to update its feature and eventually generate a "blurred" token. After this merging process, we remove $f_{v,i}^{A,m}$ from $F_{v,i}^A$, and rearrange $F_{v,i}^A$ and $F_{v,i}^B$ into updated visual features $F'_{v,i} \in \mathbb{R}^{(N_{v,i}-r) \times d_v}$ according to their original indexes. The same operations are also applied to $w_i^{A,m}$ and $w_i^{B,n}$, which are the text-aware weights of $f_{v,i}^{A,m}$ and $f_{v,i}^{B,n}$, respectively, resulting in an updated weight matrix $W'_i \in \mathbb{R}^{(N_{v,i}-r) \times 1}$ (see Eq. (7)). In this way, the proposed ToB module explicitly reduces the number of text-irrelevant background tokens, while maximally preserves those related to the target object.

Finally, ToB further enhances the visual features $F'_{v,i}$ by emphasizing its text-referred areas via $W'_i$:

$$\hat{F}_{v,i} = F'_{v,i} \odot W'_i, \tag{8}$$

where $\odot$ denotes the Hadamard product. $\hat{F}_{v,i}$ is then fed to the $(i+1)$-th transformer layer as input.

### 3.3 Transformer-Based Model with ToB

As shown in Figure 2, for addressing VG tasks, we build a transformer-based model by integrating the proposed ToB module. We adopt DINOv2-B (Oquab et al., 2023) and BERT-B (Devlin et al., 2018) as the visual and linguistic encoder, respectively. To avoid the potential misalignment between shallow visual features and textual representations, we only insert ToB modules into the last 6 transformer layers of the visual encoder. To fully exploit multi-level information of input images, we concatenate the outputs from the last $k$ visual encoder layers to form the visual features $F_v$, which are then combined with linguistic features $F_l$ and passed through a multi-modal decoder to generate the final prediction results. This multi-level feature fusion strategy enables the model to leverage both low-level spatial details and high-level semantic information, thus enhancing its ability to accurately localize objects in complex scenes. As for the decoder, we employ a multi-stage cascade architecture similar to that used in (Yang et al., 2022a). It consists of $L$ stages, where the learnable regression token [REG] iteratively aggregates useful information from $F_v$ and $F_l$ in each stage. Finally, based on the representations of [REG], a regression head predicts the bounding

box coordinates $\hat{b}_i = (\hat{x}_i, \hat{y}_i, \hat{w}_i, \hat{h}_i)$ of the text-referred object for each decoder stage. With these predictions, we define the following loss function for optimizing our transformer-based model:

$$\mathcal{L} = \sum_{i=1}^{L} [\lambda_{L1} \mathcal{L}_{L1}(\hat{b}_i, b) + \lambda_{giou} \mathcal{L}_{giou}(\hat{b}_i, b)], \tag{9}$$

where $\mathcal{L}_{L1}(\cdot, \cdot)$ and $\mathcal{L}_{giou}(\cdot, \cdot)$ represent the smooth L1 loss (Girshick, 2015) and the GIoU loss (Rezatofighi et al., 2019), respectively. $\lambda_{L1}$ and $\lambda_{giou}$ balance the relative importance of these losses. $b = (x, y, w, h)$ represents the ground-truth bounding box. During the inference phase, only $\hat{b}_L$ in the last stage is taken as the prediction result for the testing data.

## 4 EXPERIMENTS

### 4.1 EXPERIMENTAL SETTING

**Datasets**  To evaluate the effectiveness of our proposed Token Blurring module, we conduct extensive experiments and a series of ablation studies on three widely-used visual grounding datasets, including RefCOCO (Yu et al., 2016), RefCOCO+ (Yu et al., 2016), RefCOCOg (Mao et al., 2016). Their detailed descriptions are presented as follows:

**RefCOCO** (Yu et al., 2016) includes 19,994 images with 50,000 referred objects, where each image may contain multiple instances from the same object categories. There are 142,210 referring expressions in total, so each instance may get more than one text description. Following the standard setup, the image samples in RefCOCO are officially split into train/validation/testA/testB subsets that have 120,624/10,834/5,657/5,095 expressions, respectively.

**RefCOCO+** (Yu et al., 2016) consists of 19,992 images with 49,856 referred objects and 141,564 referring expressions. The usage of location-related words (e.g., "left" or "right") is strictly disallowed in the expressions from this dataset. RefCOCO+ is also officially split into train, validation, testA and testB sets with 120,191, 10,758, 5,726 and 4,889 expressions, respectively.

**RefCOCOg** (Mao et al., 2016) has 95,010 long expressions collected on Amazon Mechanical Turk for 49,856 referred objects in 25,799 images. Among them, 85,474 expression-referent pairs are selected for model training, and the remaining 9,536 expressions are separated following two different strategies, namely RefCOCOg-google (Mao et al., 2016) (val-g) and RefCOCOg-umd (Nagaraja et al., 2016) (val-u and test-u). We conduct experiments on these two partitions to make comprehensive comparisons.

**Implementation Details**  Besides our transformer-based model, the proposed ToB module can be easily integrated into other VG approaches in a plug-and-play manner. Therefore, we also verify the generalization ability of ToB by inserting it into CLIP-VG (Shi et al., 2023) and SimVG (Dai et al., 2024) with different encoder architectures. For a fair comparison, we follow the standard preprocessing protocols. The input images are resized to the resolution of $224 \times 224$, $518 \times 518$ and $640 \times 640$ for CLIP-VG, Baseline (i.e., our transformer-based model without ToB modules) and SimVG as well as their corresponding enhanced models with ToB, respectively. In addition, we also keep the same maximum expression length used in CLIP-VG and SimVG, truncating expressions that exceed this limit. Whereas for Baseline and Baseline+ToB models, the maximum token length is set to 40. The special `[CLS]` and `[SEP]` tokens are appended to the beginning and end of each expression before it is input into the linguistic encoder.

**Training Details**  Our designed transformer-based model (i.e., Baseline+ToB) is trained by using the AdamW optimizer (Loshchilov & Hutter, 2017) with a batch size of 32 and a weight decay of $10^{-4}$. Similar to (Liu et al., 2024b), we initialize the visual and linguistic encoders with the pretrained DINOv2-B and BERT-B models, respectively. Both encoders are optimized with an initial learning rate of $10^{-5}$. As for our cascade decoder, we randomly initialize its parameters by the Xavier (Glorot & Bengio, 2010) scheme and set its initial learning rate to $10^{-4}$. We train our model for 90 epochs and decrease the learning rate by a factor of 10 after 60 epochs and keep the original training strategy for other methods. For CLIP-VG+ToB and SimVG+ToB, we directly integrate the ToB modules into CLIP-VG and SimVG, respectively, while maintaining their architectures and initializations. For a fair comparison, we utilize the same training strategy as the original methods. In addition, we also adopt the data augmentation strategy used in the previous works (Yang et al.,

| Methods | Visual Encoder | Linguistic Encoder | RefCOCO | | | RefCOCO+ | | | RefCOCOg | | |
|---|---|---|---|---|---|---|---|---|---|---|---|
| | | | val | testA | testB | val | testA | testB | val-g | val-u | test-u |
| **Pretrained on close-set detection and natural language processing tasks:** | | | | | | | | | | | |
| TransVG (Deng et al., 2021) | ResNet-101 | BERT-B | 81.02 | 82.72 | 78.35 | 64.82 | 70.70 | 56.94 | 67.02 | 68.67 | 67.73 |
| LUNA (Liang et al., 2023) | ResNet-101 | BERT-B | 84.67 | 86.74 | 80.21 | 72.79 | 77.98 | 64.61 | - | 74.16 | 72.85 |
| VLTVG (Yang et al., 2022a) | ResNet-101 | BERT-B | 84.77 | 87.24 | 80.49 | 74.19 | 78.93 | 65.17 | 72.98 | 76.04 | 74.18 |
| MMCA (Yao et al., 2024) | ResNet-101 | BERT-B | 84.76 | 87.34 | 80.86 | 73.18 | 78.67 | 64.13 | 72.53 | 74.91 | 73.87 |
| PBREC-MT (Zhao et al., 2024) | ResNet-101 | BERT-B | 82.94 | 86.31 | 80.81 | 74.85 | 79.53 | 65.60 | - | 73.86 | 74.13 |
| SegVG (Kang et al., 2024) | ResNet-101 | BERT-B | 86.84 | 89.46 | 83.07 | 77.18 | 82.63 | 67.59 | 76.01 | 78.35 | 77.42 |
| AttBalance (Kang et al., 2025a) | ResNet-101 | BERT-B | 85.30 | 88.13 | 81.50 | 75.14 | 80.25 | 66.34 | 74.08 | 77.35 | 75.61 |
| TransVG++ (Deng et al., 2023) | ViT-B | BERT-B | 86.28 | 88.37 | 80.97 | 75.39 | 80.45 | 66.28 | 73.86 | 76.18 | 76.30 |
| VG-LAW Su et al. (2023) | ViT-B | BERT-B | 86.06 | 88.56 | 82.87 | 75.74 | 80.32 | 66.69 | - | 75.61 | 76.28 |
| QRNet (Ye et al., 2022) | Swin-S | BERT-B | 84.01 | 85.85 | 82.34 | 72.94 | 76.17 | 63.81 | 71.89 | 73.03 | 72.52 |
| VG-LAW (Su et al., 2023) | Swin-S | BERT-B | 84.82 | 87.22 | 81.94 | 74.36 | 78.49 | 65.24 | - | 75.61 | 76.28 |
| PVD (Cheng et al., 2024) | Swin-B | BERT-B | 84.99 | 88.02 | 80.03 | 74.27 | 79.06 | 65.11 | 74.34 | 74.64 | 71.41 |
| **Pretrained on large-scale datasets with generative vision-language models:** | | | | | | | | | | | |
| Ferret-7B (You et al., 2024) | CLIP-L | Vicuna-7B | 87.49 | 91.35 | 82.45 | 80.78 | 87.38 | 73.14 | - | 83.93 | 84.76 |
| ExpVG (Kang et al., 2025b) | CLIP-L | Vicuna-7B | 87.40 | 91.70 | 81.50 | 80.30 | 86.90 | 71.10 | 88.00 | 81.30 | 81.40 |
| Groma-7B (Ma et al., 2024) | DINOv2-L | Vicuna-7B | 89.35 | 92.09 | 86.26 | 83.90 | 88.91 | 78.05 | - | 86.37 | 87.01 |
| MoVA-7B (Zong et al., 2024) | Multi-experts | Vicuna-7B | 92.55 | 94.50 | 88.81 | 87.70 | 92.05 | 82.94 | - | 89.28 | 89.70 |
| Qwen-VL-7B (Bai et al., 2023) | ViT-bigG | Qwen-7B | 89.36 | 92.26 | 85.34 | 83.12 | 88.25 | 77.21 | - | 85.58 | 85.48 |
| Qwen2.5-VL-7B (Bai et al., 2025) | FE-ViT | Qwen2.5-7B | 90.00 | 92.50 | 85.40 | 84.20 | 89.10 | 76.90 | - | 87.20 | 87.20 |
| InternVL3-8B (Zhu et al., 2025) | InternViT-300M | Qwen2.5-7B | 92.50 | 94.60 | 88.00 | 88.20 | 92.50 | 81.80 | - | 89.60 | 90.00 |
| **Pretrained on large-scale vision / vision-language tasks through self-supervised learning:** | | | | | | | | | | | |
| CLIP-VG (Shi et al., 2023) | CLIP-B | CLIP-B | 84.29 | 87.76 | 78.43 | 69.55 | 77.33 | 57.62 | 72.64 | 73.18 | 72.54 |
| CLIP-VG + ToB (Ours) | CLIP-B | CLIP-B | 86.03 | 88.47 | 80.50 | 75.23 | 82.11 | 65.36 | 75.14 | 78.45 | 77.88 |
| Performance Improvement △ | - | - | +1.74 | +0.71 | +2.07 | +5.68 | +4.78 | +7.74 | +2.50 | +5.27 | +5.34 |
| D-MDETR (Shi et al., 2023) | CLIP-B | CLIP-B | 85.97 | 88.82 | 80.12 | 74.83 | 81.70 | 63.44 | 72.21 | 74.14 | 74.49 |
| RefFormer (Wang et al., 2024) | CLIP-B | CLIP-B | 86.52 | 90.24 | 81.42 | 76.58 | 83.69 | 67.38 | - | 77.80 | 77.60 |
| TransVG (Deng et al., 2021) | DINOv2-B | BERT-B | 85.31 | 87.80 | 81.03 | 74.57 | 80.22 | 65.31 | 73.76 | 74.22 | 75.02 |
| MaPPER (Liu et al., 2024b) | DINOv2-B | BERT-B | 86.03 | 88.90 | 81.19 | 74.92 | 81.12 | 65.68 | 74.60 | 76.32 | 75.81 |
| Baseline (Ours) | DINOv2-B | BERT-B | 85.55 | 87.04 | 80.57 | 75.99 | 80.30 | 66.54 | 74.97 | 76.67 | 76.05 |
| Baseline + ToB (Ours) | DINOv2-B | BERT-B | 87.73 | 89.66 | 83.91 | 78.73 | 83.37 | 68.81 | 76.89 | 80.39 | 80.36 |
| Performance Improvement △ | - | - | +2.18 | +2.62 | +3.34 | +2.74 | +3.07 | +2.27 | +1.92 | +3.74 | +4.31 |
| SimVG (Dai et al., 2024) | BEiT-3 | BEiT-3 | 87.63 | 90.22 | 84.04 | 78.65 | 83.36 | 71.82 | 78.81 | 80.37 | 80.51 |
| SimVG + ToB | BEiT-3 | BEiT-3 | 88.95 | 91.47 | 84.87 | 80.02 | 85.45 | 73.11 | 80.03 | 82.23 | 81.79 |
| Performance Improvement △ | - | - | +1.32 | +1.25 | +0.83 | +1.37 | +2.09 | +1.29 | +1.22 | +1.86 | +1.28 |
| **Pretrained on visual grounding tasks with multiple/extra datasets:** | | | | | | | | | | | |
| RefFormer (Wang et al., 2024) | CLIP-B | CLIP-B | 88.82 | 92.52 | 84.87 | 80.91 | 86.64 | 73.35 | - | 82.29 | 83.15 |
| HiVG (Xiao et al., 2024b) | CLIP-B | CLIP-B | 90.56 | 92.55 | 87.23 | 83.08 | 89.21 | 76.68 | - | 84.52 | 85.62 |
| EEVG(Chen et al., 2024) | ViT-B | BERT-B | 90.47 | 92.73 | 87.72 | 81.79 | 87.80 | 74.94 | - | 85.19 | 84.72 |
| SimVG (28k) (Dai et al., 2024) | BEiT-3 | BEiT-3 | 90.98 | 92.68 | 87.94 | 84.17 | 88.58 | 78.53 | - | 85.90 | 86.23 |
| SimVG (28k) + ToB | BEiT-3 | BEiT-3 | 91.20 | 93.11 | 87.43 | 84.67 | 89.31 | 78.89 | - | 86.85 | 86.77 |
| Performance Improvement △ | - | - | +0.22 | +0.34 | -0.51 | +0.50 | +0.73 | +0.36 | - | +0.95 | +0.54 |

Table 1: Performance comparison with state-of-the-art methods on the RefCOCO, RefCOCO+ and RefCOCOg datasets. The performance of three main benchmarks and their corresponding ToB-enhanced models are shaded in red and blue , respectively.

2019b; 2020; Deng et al., 2021; Ye et al., 2022) during the training phase. The common space dimension value $d_c$ is set to $d_c = 256$. For the loss function defined in Eq. (9), the hyperparameters are as $\lambda_{L1} = 5$, $\lambda_{giou} = 2$ and the number of decoder stages $L$ is set to $L = 6$.

**Evaluation Metric** The standard ACC@0.5 protocol (Yang et al., 2019b; 2020; Ye et al., 2022) is used to evaluate the performance of different competing methods for solving VG tasks. Specifically, given an image-expression pair as input, the prediction is considered to be correct only if the IoU value between it and the corresponding ground-truth bounding box is greater than 0.5.

## 4.2 MAIN RESULTS

**Comparison with State-of-the-art Methods** Table 1 compares the results of our method and state-of-the-art VG approaches on RefCOCO, RefCOCO+, and RefCOCOg. From these results, we get the following observations: (1) Without pre-training on extra datasets, our Baseline+ToB model outperforms all other competing methods. Its improvements over MaPPER using the same DINOv2-B/BERT-B encoders range from 0.76% (89.66% vs. 88.90% on RefCOCO testA set) to 4.55% (80.36% vs. 75.81% on RefCOCOg test-u set). By inserting ToB modules into SimVG(28k), which is pre-trained on a combined dataset of 28K samples, SimVG(28k)+ToB model achieves comparable results to various large vision-language models. Specifically, it consistently outperforms Ferret-7B, Groma-7B, and Qwen-VL-7B across all datasets, and is competitive with Qwen2.5-VL-7B on most cases. These results demonstrate the effectiveness of our proposed ToB module. (2) Compared to the original CLIP-VG, Baseline, and SimVG, the corresponding enhanced models generally exhibit

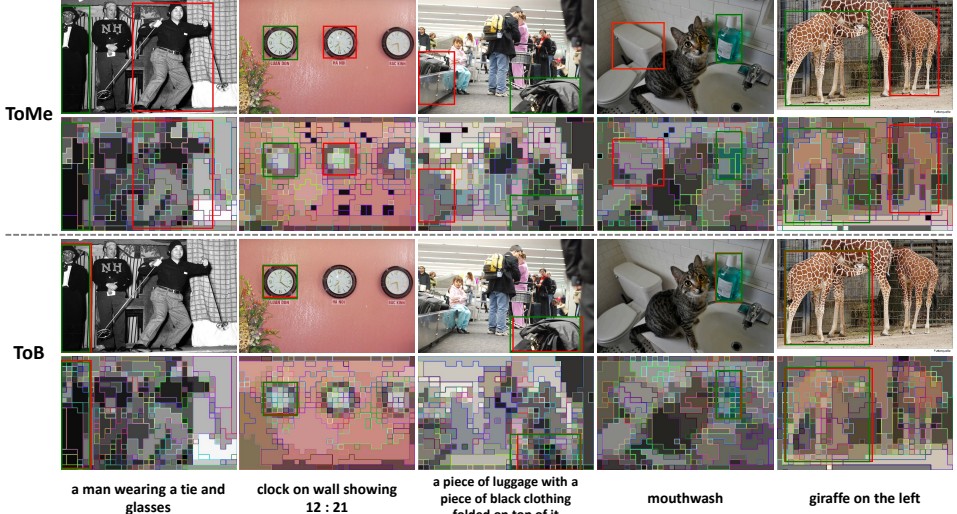

ToMe

ToB

a man wearing a tie and glasses

clock on wall showing 12 : 21

a piece of luggage with a piece of black clothing folded on top of it

mouthwash

giraffe on the left

Figure 3: Visualizations of token merging results in the last visual encoder layer, which are obtained by iterating ToMe and our ToB modules. The ground-truth bounding boxes and prediction results are shown in green and red colors.

much better performance. Especially for CLIP-VG, ToB brings improvement by at least 0.71% (88.47% vs. 87.76% on RefCOCO testA set) and up to 7.74% (65.36% vs. 57.62% on RefCOCO+ testB set). This suggests that the VG performance can be improved by merging text-irrelevant tokens as in ToB, and also verifies the generalization ability of ToB across different model architectures.

**Comparison with Existing Token Merging Methods** We further compare our ToB method with existing token merging strategies under the same token reduction rate, including ToMe (Bolya & Hoffman, 2023) and ToE (Huang et al., 2024). As shown in Table 2, integrating ToMe and ToE improves the model efficiency compared to Baseline, increasing the inference speed from 20.8 fps to 22.09 fps and 22.93 fps, respectively. However, both of them suffer from a performance degradation. It is

| Methods | RefCOCOg | | fps |
|---|---|---|---|
| | val-u | test-u | |
| Baseline | 76.67 | 76.05 | 20.80 |
| Baseline + ToMe | 76.41 (-0.26) | 75.25 (-0.80) | 22.09 |
| Baseline + ToE | 75.13 (-1.54) | 74.89 (-1.16) | 22.93 |
| Baseline + ToB | **80.39 (+3.72)** | **80.36 (+4.31)** | 21.57 |

Table 2: Effectiveness and efficiency comparison with other token merging methods.

mainly because selecting tokens solely based on visual similarity can not accurately localize the background tokens that are irrelevant to the input expression, so some critical object-related tokens will be merged and such information loss may decrease the final detection accuracy. In contrast, by introducing text relevance as an additional guidance information, Baseline+ToB achieves superior performance, 80.39% on val-u split and 80.36% on test-u split, surpassing Baseline by 3.72% and 4.31%, respectively. *These results highlight the advantage of incorporating text relevance in token merging process and the importance of correctly identifying background tokens for VG tasks.* In addition, by reducing the number of image tokens, ToB also increases the inference speed of Baseline to 21.57 fps. This further demonstrates that our ToB module can effectively reduce the computational overhead.

To better understand the impact of different token merging strategies on VG performance, we visualize the merged token maps and detection results achieved by Baseline+ToMe and Baseline+ToB in Figure 3. As can be seen, since ToMe cannot identify background tokens, it merges a lot of tokens within the target areas (see green boxes), thus losing the details information of the target object. By contrast, the merged tokens of ToB are generally located in the text-irrelevant areas, while the target-related foreground tokens are mostly preserved. For the query "giraffe on the left", ToB produces finer token groupings around target giraffe, yielding precise left-right disambiguation, whereas ToMe merges both giraffes into a single coarse region. Overall, these visualization results confirm that ToB's language-guided merging strategy enables better preservation of semantically

| Fusion layer $k$ | 1 | 2 | 3 | 4 | 5 |
|---|---|---|---|---|---|
| RefCOCOg (val-u) | 79.71 | 80.27 | **80.39** | 80.25 | 79.66 |
| RefCOCOg (test-u) | 79.69 | 80.14 | **80.36** | 80.04 | 79.25 |

Table 3: Ablation study on the number of fusion layers $k$.

| Blurring Tokens $r$ | 0 | 4 | 16 | 32 | 64 | 96 | 128 |
|---|---|---|---|---|---|---|---|
| RefCOCOg (val-u) | 78.91 | 79.45 | 80.05 | 80.17 | **80.39** | 79.93 | 79.26 |
| RefCOCOg (test-u) | 79.12 | 79.61 | 79.78 | 79.64 | **80.36** | 79.87 | 78.84 |
| fps | 20.55 | 20.64 | 20.88 | 21.09 | 21.57 | 21.76 | 22.05 |

Table 4: Ablation study on the number of blurring tokens $r$.

relevant tokens and successful suppression of text-irrelevant features, resulting in more accurate and efficient grounding results.

### 4.3 ABLATION STUDIES

We conduct ablation experiments to study the effects of different hyperparameters in our ToB module. All experiments are performed on the RefCOCOg dataset under with same training strategy.

**Effect of Fusion Layer Number**  As mentioned in previous section, we concatenate the outputs from the last $k$ layers of the visual encoder to fully exploit the multi-level information of input images. In this subsection, we investigate the effect of the number of layers used in this multi-level feature fusion strategy, by comparing the model performance with different values of $k$. As shown in Table 3, our model achieves the best results when $k = 3$, outperforming those at $k = 1$. This indicates that some valuable information may be lost due to the excessive token merging process in the deep layers, which can be compensated by the multi-level feature fusion process. While when $k$ exceeds 3, there are still large number of background tokens contained in the visual features, which can introduce noise that negatively influences the model performance. These findings demonstrate that appropriately adopting multi-level features can alleviate the issue of ToB and further promote its effectiveness. Therefore, we set $k = 3$ for our transformer-based model.

**Effect of Blurring Token Number**  To investigate the impact of token reduction on performance and efficiency, we conduct an ablation study by varying the number of merged tokens $r$ in our ToB module. As shown in Table 4, increasing $r$ from 0 to 64 consistently improves both grounding accuracy and inference speed. Specifically, performance rises from 78.91% to 80.39% on the val-u split and from 79.12% to 80.36% on the test-u split, with a corresponding fps gain from 20.55 to 21.57. This suggests that moderate token merging effectively reduces redundancy while preserving or even enhancing critical visual-textual information. However, further increasing $r$ beyond 64 leads to marginal or negative gains. At $r = 96$, performance slightly drops on both splits, and at $r = 128$, the accuracy degrades more noticeably to 79.26% (val-u) and 78.84% (test-u), despite achieving the highest speed (22.05 fps). These results indicate that excessive merging may lead to the loss of fine-grained features essential for precise localization. Based on these analyzes, we choose $r = 64$ in our model for the best trade-off between model performance and efficiency.

**Efficiency Improvement Analysis**  Table 5 reports a detailed efficiency analysis under different token merging strategies, in terms of computational overhead (GFLOPs), memory usage (GB), and inference speed (fps). We can observe a trade-off between enhancing model performance and improving its efficiency. Blurring more tokens in the last six layers of the visual encoder yields greater efficiency gains but decreases the grounding accuracy. More specifically, DINOv2-B takes images with a resolution of $518 \times 518$ as input, and divides them into $14 \times 14$-sized patches, producing 1369 tokens per layer. When the blurring token number $r$ in each ToB module increases from 64 to 196, the amount of tokens remaining in the final visual encoder layer plummets from $1369 - 6 \times 64 = 985$ to $1369 - 6 \times 196 = 193$, thus resulting in a more significant efficiency improvement over the Baseline model (FLOPs decrease from 118.31 G to 99.42 G, memory usage reduces from 19.97 GB to 16.84 GB, and fps increases from 20.80 to 23.96). Although $r = 196$ leads to a performance degradation compared to $r = 64$, the Baseline+ToB under different settings still consistently outperforms Baseline by at least 2.24% (78.91% vs. 76.67%) and 2.72% (78.77% vs. 76.05%) on the two splits,

| Methods | $r$ | RefCOCOg ↑ val-u | test-u | GFLOPs ↓ | Memory (GB) ↓ | fps ↑ |
|---|---|---|---|---|---|---|
| Baseline | - | 76.67 | 76.05 | 118.31 | 19.97 | 20.80 |
| Baseline + ToB | 64 | 80.39 | 80.36 | 113.85 | 19.21 | 21.57 |
| Baseline + ToB | 128 | 79.26 | 78.84 | 107.96 | 18.18 | 22.05 |
| Baseline + ToB | 196 | 78.91 | 78.77 | 99.42 | 16.87 | 23.96 |
| ToMe + ToB | 96 | 77.53 | 77.07 | 76.84 | 14.78 | 28.71 |

Table 5: Detailed efficiency analysis with different implementation strategies.

| Method | RefCOCOg | Accuracy @ IoU | | | | | Scale-wise Accuracy @0.5 | | |
|---|---|---|---|---|---|---|---|---|---|
| | | @0.5 | @0.6 | @0.7 | @0.8 | @0.9 | $ACC_s$ | $ACC_m$ | $ACC_l$ |
| Baseline (Ours) | val-u | 76.67 | 72.59 | 66.47 | 54.34 | 24.83 | 60.59 | 74.11 | 85.49 |
| Baseline + ToB ($r = 64$) | val-u | 80.39 | 77.25 | 72.81 | 65.22 | 47.49 | 67.27 | 78.47 | 88.43 |
| Baseline + ToB ($r = 196$) | val-u | 78.91 | 77.13 | 72.43 | 64.41 | 46.10 | 67.48 | 77.33 | 86.05 |
| Baseline (Ours) | test-u | 76.05 | 70.55 | 61.33 | 49.81 | 26.92 | 65.32 | 73.79 | 83.22 |
| Baseline + ToB ($r = 64$) | test-u | 80.36 | 78.01 | 73.24 | 66.82 | 48.38 | 72.03 | 79.21 | 85.08 |
| Baseline + ToB ($r = 196$) | test-u | 78.77 | 76.96 | 72.55 | 65.07 | 46.54 | 71.64 | 77.40 | 83.65 |

Table 6: Fine-grained evaluation results on RefCOCOg.

respectively, also surpassing TransVG and MaPPER employing the same encoders (see Table 1). These results demonstrate the effectiveness and efficiency of our proposed ToB module.

Moreover, we design a ToMe+ToB configuration by integrating ToMe into Baseline+ToB to further compress tokens in the first six visual encoder layers, enabling aggressive token reduction across all twelve layers. By setting $r = 96$, ToMe+ToB achieves both higher accuracy than Baseline and the best model efficiency (76.84 GFLOPs, 14.78 GB memory, and 28.71 fps) among all test variants. This indicates that our ToB module is compatible with other token compression strategies and can serve as a strong, complementary module for efficient visual grounding.

**Fine-grained Grounding Performance**   To better assess the effect of ToB on localization precision, Table 6 presents the grounding accuracy achieved under stricter IoU thresholds and across different object scales. Compared to the Baseline model, ToB consistently brings improvements across all IoU thresholds. When $r = 64$, the performance on the val-u split increases from 76.67% to 80.39% at 0.5 IoU, and such gains become even more pronounced at higher thresholds (e.g., from 54.34% to 65.22% at 0.8 IoU and from 24.83% to 47.49% at 0.9 IoU). Similar trends can also be observed on the test-u split and under the $r = 196$ setting, suggesting that suppressing text-irrelevant background tokens helps the model maintain more accurate visual-textual alignment.

Regarding the scale-wise evaluation, the target objects are divided into three groups, namely small ($< 128 \times 128$ pixels), medium ($128 \times 128$–$256 \times 256$ pixels), and large ($> 256 \times 256$ pixels), based on their bounding-box area. Notably, ToB yields substantial improvements, particularly on small objects where background clutter tends to dominate the overall token distribution. All the results mentioned above confirm that our ToB can preserve the fine-grained spatial information required for high-precise bounding box regression.

## 5 CONCLUSION

In this paper, we propose Token Blurring (ToB), a novel language-guided token merging module to improve the effectiveness and efficiency of solving visual grounding tasks. Unlike previous approaches that either enhance attention on foreground objects or reduce token redundancy solely based on visual similarity, ToB jointly considers visual similarity and textual relevance to selectively blur redundant and text-irrelevant background tokens. When integrated into a transformer-based framework, extensive experiments demonstrate that ToB enables more accurate localization by explicitly suppressing distractive text-irrelevant background features while preserving critical visual-linguistic correlation information.

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

# A  APPENDIX

## A.1  MORE ABLATION STUDIES

**Effect of Balance Between Visual Similarity and Text Relevance**   In order to accurately identify background tokens from the input image, we introduce a positive parameter $\alpha$ to balance the importance between visual similarity and text relevance criteria (see Eq. (5) in the main text). In this subsection, we analyze the impact of $\alpha$ on the performance of our proposed ToB module by varying its value from 0.0 to 1.0 with an interval of 0.1, as shown in Table 7. As can be seen, when $\alpha$ is set to 0, ToB relies solely on visual similarity for background token identification, achieving the detection accuracy of 79.37% and 78.91% on val-u and test-u splits, respectively. This suggests that overlooking the text relevance may result in the loss of essential information for addressing VG tasks during the token merging procedure. As $\alpha$ increasing, the model performance improves gradually due to the integration of text relevance guidance, which reaches peak values of 80.39% on val-u and 80.36% on test-u at $\alpha = 0.3$, clearly surpassing the baseline scenario (i.e., $\alpha = 0.0$). When $\alpha$ exceeds 0.3, our ToB module becomes overly reliant on the text relevance. It will merge some tokens with dissimilar visual features, which may be belonging to different areas with distinct semantics. As a result, some important context features for correctly localizing the target object will be lost, thus reducing the the model performance progressively. The above results indicate that maintaining a proper balance between visual similarity and text relevance is critical for the success of our ToB module. We thus fix $\alpha = 0.3$ for all experiments in this paper.

| Balance Parameter $\alpha$ | 0.0 | 0.1 | 0.2 | 0.3 | 0.4 | 0.5 |
|---|---|---|---|---|---|---|
| RefCOCOg (val-u) | 79.37 | 79.53 | 79.97 | **80.39** | 79.86 | 79.57 |
| RefCOCOg (test-u) | 78.91 | 78.95 | 79.61 | **80.36** | 79.62 | 79.51 |
| Balance Parameter $\alpha$ | 0.6 | 0.7 | 0.8 | 0.9 | 1.0 | - |
| RefCOCOg (val-u) | 79.35 | 79.45 | 79.13 | 78.97 | 78.92 | - |
| RefCOCOg (test-u) | 78.91 | 78.92 | 78.66 | 78.54 | 78.40 | - |

Table 7: Ablation study on the balance parameter $\alpha$.

**Effect of Further Feature Enhancement**   In Eq. (8) in the main text, ToB further enhances the merged visual features with the updated text-aware weights. By setting $\alpha$ in Eq. (5) to 0, our model degrades into a Baseline+ToMe model equipped with this feature enhancement operation. Therefore, by comparing the results obtained at $\alpha = 0.0$ in Table 7 with those of Baseline+ToMe in Table 2 in the main text, we can find that this degraded ToB module significantly outperforms Baseline+ToMe by 2.96% (79.37% vs. 76.41%) on val-u and 3.66% (78.91% vs. 75.25%) on test-u, demonstrating the effectiveness and importance of the feature enhancement operation for our proposed ToB module.

**Effect of Aggregation Methods**   Table 8 compares different aggregation strategies used in Eq. (3) for computing the text-aware weights. The average pooling operation $\mathcal{F}_{\text{AvgPool}}(\cdot)$ is replaced with the max pooling $\mathcal{F}_{\text{MaxPool}}(\cdot)$ and the attention-weighted pooling $\mathcal{F}_{\text{AttPool}}(\cdot)$, respectively. For $\mathcal{F}_{\text{AttPool}}(\cdot)$, the attention weights are generated from the corresponding word embedding through a linear layer. As can be seen, the average pooling adopted in our ToB module achieves the best performance (80.39% on val-u and 80.36% on test-u), indicating that aggregating correlations across all linguistic tokens provides a stable and reliable signal for distinguishing foreground-relevant tokens from background ones. Nevertheless, the overall differences between different pooling strategies are relatively small, indicating that ToB is insensitive to diverse choices of aggregation methods and different pooling operations can be flexibly adopted based on practical requirements.

| Token Blurring | Average pooling | Max pooling | Attention-weighted pooling |
|---|---|---|---|
| RefCOCOg (val-u) | 80.39 | 80.09 | 80.15 |
| RefCOCOg (test-u) | 80.36 | 79.87 | 79.71 |

Table 8: Ablation study on different aggregation methods for computing text-aware weights.

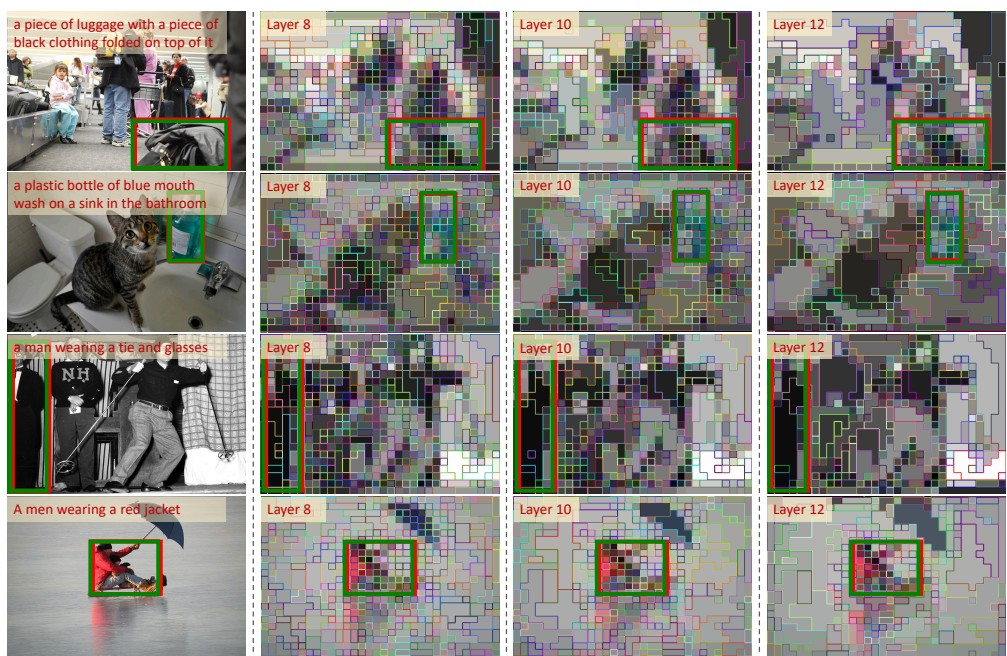

Figure 4: Visualizations of token merging results obtained at layers 8, 10, 12 in our Baseline+ToB model. The ground-truth bounding boxes and final prediction results are shown in green and red colors.

## A.2 MORE QUALITATIVE RESULTS

To understand the overall token blurring process throughout the transformer-based model, we visualize the merged token maps obtained at layers 8, 10, 12 in our Baseline+ToB model in Figure 4. We can observe that ToB gradually merges the tokens, which are more visually similar and less relevant to the input expression. For example, the cat and toilet tokens second row images, as well as the ground and umbrella tokens in fourth row images, are more likely to be merged. In contrast, the tokens located in the object-related foreground areas (see green boxes) are generally preserved. Even in the images of the third row, the merged tokens of "a man" are concentrated in the shirt and trouser areas, while those of "a tie" and "glasses" are maintained, so this will not affect the final detection result of "a man wearing a tie and glasses".

Furthermore, we visualize the text-aware weights $W_i$ obtained at layers 7, 8, 9, 10 to validate the visual-textual alignment ability of our ToB module. As shown in Figure 5, most foreground visual tokens receive noticeably higher text-aware weights than background ones. Additionally, for the same elephant image in the last two rows, the weight maps produced by different referring expressions emphasize different target instances, demonstrating that the ToB-generated weights can adapt appropriately to the semantics of the input text and guide the model toward the correct object.

We also provide additional token merging visualizations in Figure 6. In the scenes containing multiple similar objects, the text-guided nature of ToB ensures that token merging does not rely solely on the visual similarity. Instead, ToB selectively preserves rich foreground information relevant to the referring expression, even under aggressive merging settings (e.g., $r = 196$). This demonstrates that ToB can retain task-critical semantic details while still achieving substantial token reduction.

Based on the above analysis, all these results can support our claim that *our proposed ToB module has the ability to "blur" the background regions of input images by gradually merging the text-irrelevant tokens, while maximally preserving the density of text-referred foreground tokens.*

## A.3 THE USE OF LARGE LANGUAGE MODELS (LLMS)

We have not used Large Language Models (LLMs) for our paper writing.

**Expression: the torso of a brown furry teddy bear wearing a red ribbon**

**Expression: the woman who is wearing a blue shirt**

**Expression: a man dressed in all black with a black backpack**

**Expression: the elephant who has it ' s trunk curving upwards**

**Expression: the elephant behind the front one**

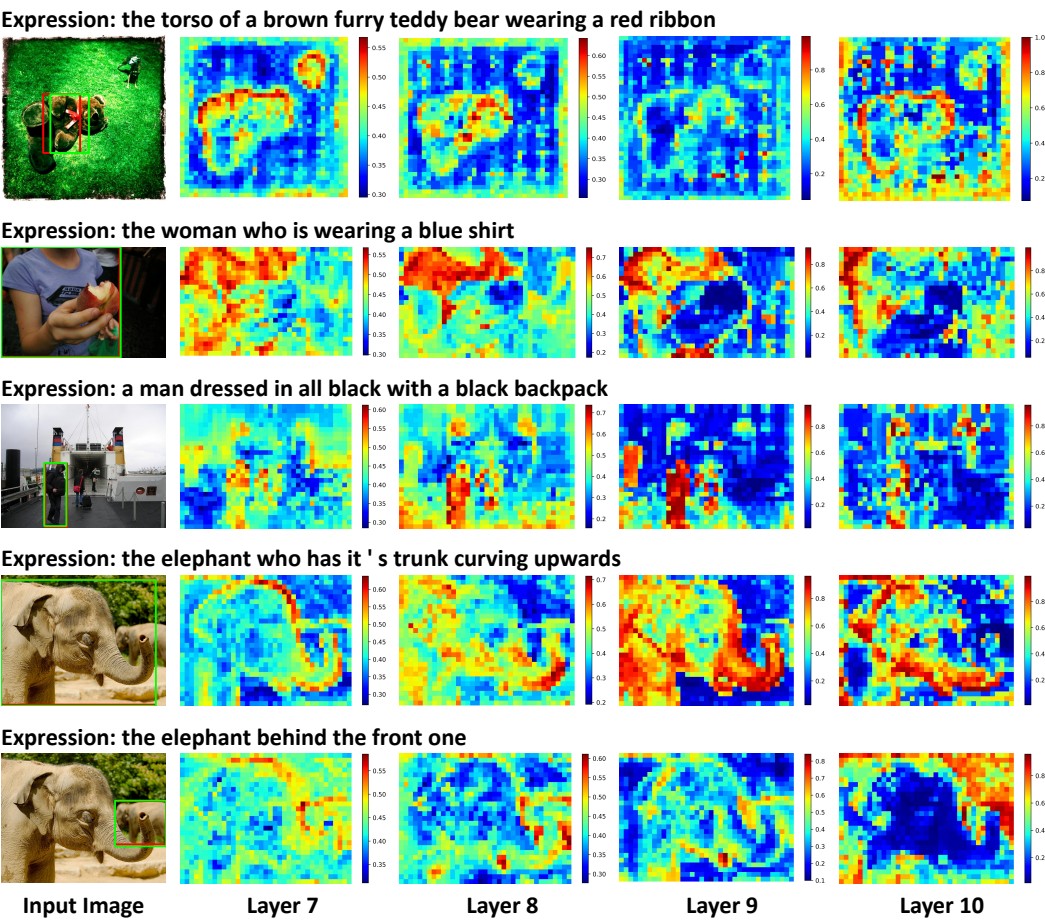

**Input Image**    **Layer 7**    **Layer 8**    **Layer 9**    **Layer 10**

Figure 5: Visualizations of text-aware weights $W_i$ obtained at layers 7, 8, 9, 10 in our Baseline+ToB model. The ground-truth bounding boxes and final prediction results are shown in red and green colors.

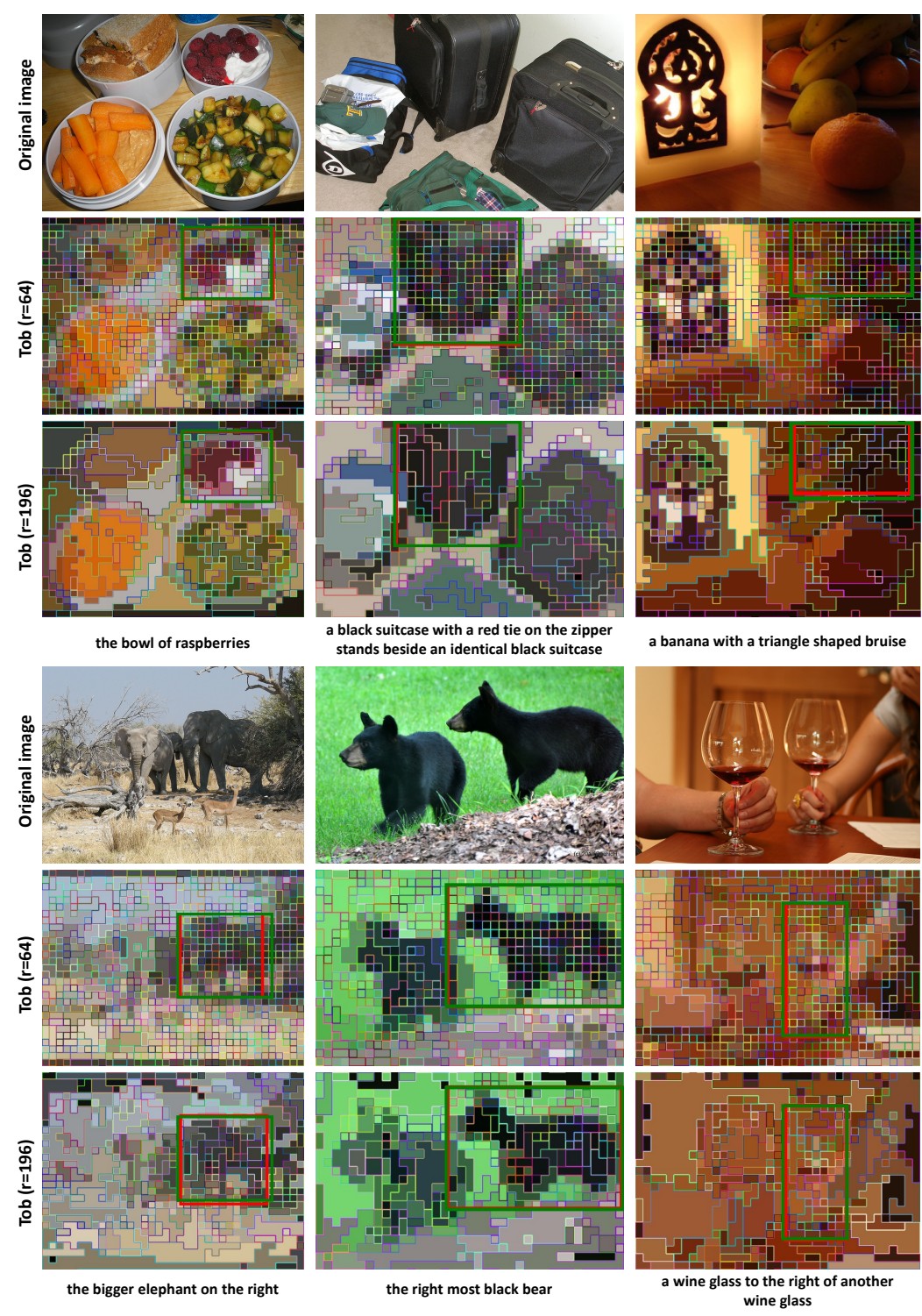

Figure 6: More visualizations of token merging results obtained at last layer in our Baseline+ToB model with different $r$ for images with multiple similar objects. The ground-truth bounding boxes and final prediction results are shown in green and red colors.

