# OpenReview forum: "Background Blurring Matters: Improving Visual Grounding by Merging Text-Irrelevant Tokens"
_ICLR.cc/2026/Conference — Submitted to ICLR 2026_

### Official Review · Reviewer_weHD · 2025-10-28

**Soundness:** 3
**Presentation:** 3
**Contribution:** 3
**Rating:** 6
**Confidence:** 5

**Summary:**

This paper introduces Token Blurring (ToB), a novel module for visual grounding (VG) that reduces computational overhead and improves accuracy by merging text-irrelevant background tokens. ToB uses a language-guided merging strategy that considers both visual similarity and textual relevance, making it more effective than prior token merging methods. Integrated into a transformer-based model (DINOv2-B + BERT-B), ToB improves grounding performance across RefCOCO, RefCOCO+, and RefCOCOg datasets, and generalizes well to other VG architectures like CLIP-VG and SimVG.

**Strengths:**

1. First to jointly use visual similarity and textual relevance for token merging in VG.
2. Unlike ToMe or ToE, ToB explicitly avoids merging foreground tokens, preserving object details.
3. Extensive experiments across 3 datasets and 3 model backbones.
4. Plug-and-play integration into existing models (CLIP-VG, SimVG) shows consistent gains (up to +7.74% on RefCOCO+ testB).

**Weaknesses:**

1. Token merging (ToMe, ToE) and task-aware pruning (LAPS, MustDrop) already exist. ToB’s core idea is incremental, not radical.
2. Fps improvement is < 1 frame (20.8 → 21.57). Not a major speedup.

**Questions:**

See Weakness

---

> ### Author Response · Authors · 2025-11-24
> **Rebuttal for Reviewer weHD**
>
> Dear Reviewer weHD, thank you for your constructive comments. We address each weakness (W) and question (Q) below.
> ***
> **W1:** Token merging (ToMe, ToE) and task-aware pruning (LAPS, MustDrop) already exist. ToB’s core idea is incremental, not radical.
>
> **R1:** To the best of our knowledge, the proposed ToB method is the first attempt to utilize both visual and textual information for token merging. It is also an innovative merging strategy designed to **compress text-irrelevant background tokens** for addressing VG tasks. In **Table 2** of our paper, we have compared ToB with the ToME and ToE methods by separately inserting them into the Baseline model. Both of these two methods select tokens solely based on visual similarity, they can not accurately localize the background tokens that are irrelevant to the input expression, so they suffer from a performance degradation compared to Baseline. In contrast, Baseline+ToB achieves superior performance, 80.39% on val-u split and 80.36% on test-u split, surpassing Baseline by 3.72% and 4.31%, respectively. **These results highlight the advantage of incorporating text relevance in the token merging process and the importance of correctly identifying background tokens for VG tasks.** Therefore, we believe that our ToB method is a novel technique that can bring significant improvements in grounding accuracy and can be easily integrated into other VG approaches in a plug-and-play manner.
> ***
> **W2:** Fps improvement is < 1 frame (20.8 → 21.57). Not a major speedup.
>
> **R2:** To answer this question, we have added a detailed efficiency analysis in **Table 5** of our revised manuscript. As can be seen, our ToB modules consistently enhance the overall efficiency of the Baseline model across different implementation settings, in terms of computational overhead (GFLOPs), memory usage (GB), and inference speed (fps). **Such improvements become even more pronounced when the number of blurring tokens $r$ in each ToB module increases from 64 to 196**. When $r=196$, Baseline+ToB realizes significant efficiency improvements over Baseline by **18.89 GFLOPs, 3.13GB memory usage, and 3.14 fps.** Although the grounding accuracy of Baseline+ToB drops as $r$ increases, the worst-performed variant achieved at $r=196$ still outperforms the Baseline model by 2.24% (78.91% vs. 76.67%) and 2.72% (78.77% vs. 76.05%) on the two splits, respectively, and also surpasses TransVG and MaPPER employing the same encoders (see **Table 1**). This demonstrates that integrating our ToB modules can bring benefits to both the efficiency and effectiveness of VG models.

---

> ### Author Response · Authors · 2025-11-24
>
> Dear Reviewer weHD,
>
> Thanks for your reviews. We've tried our best to provide our responses and revise our paper. Could you please let us know if you need any further clarifications or discussions?
>
> We would really appreciate it if we could get your feedback.
>
> Best regards,
>
> Authors.

---

> > ### Comment · Reviewer_weHD · 2025-11-27
> >
> > Thanks for the response. The response addresses my concerns. I will keep my score.

---

### Official Review · Reviewer_Fusm · 2025-10-31

**Soundness:** 2
**Presentation:** 3
**Contribution:** 2
**Rating:** 4
**Confidence:** 5

**Summary:**

This paper studies visual grounding. The authors point out that current transformer-based VG models are affected by many text-irrelevant background tokens, which not only introduce noise in attention but also increase computational cost. To solve this, they propose Token Blurring (ToB), a module that merges image tokens based on both visual similarity and textual relevance. Different from previous token merging methods like ToMe that only consider visual similarity, ToB assigns text-aware weights and merges visually similar but text-irrelevant tokens while keeping important foreground ones. When added to models such as CLIP-VG and a DINOv2-B/BERT-B baseline, ToB consistently improves accuracy on RefCOCO, RefCOCO+, and RefCOCOg with faster inference.

**Strengths:**

1. Motivation is clear. The paper clearly identifies an issue with redundant background tokens and proposes a neat and intuitive fix. Making token merging text-aware is a simple but effective idea.

2. It's a plug-and-play module. ToB works across multiple backbones and datasets, showing that it is not tied to a single architecture. I think this is quite a practical contribution. But there's also a concern regarding this perspective. I articulate it in the weakness.

3. The authors test many variants and provide clear visualizations, which help me understand how ToB preserves important foreground regions compared to baselines like ToMe.

**Weaknesses:**

1. The paper misses a lot of recent works on Visual Grounding. Therefore, the related work section is seriously incomprehensive. Tables should cite and compare SegVG (ECCV'24), AttBalance (ACM-MM), ExpVG, etc. Those are all visual grounding methods without comparison and discussion in this paper.

2. Textual relevance computation seems oversimplified. In Eq. (3), you just take the average correlation between image and language tokens. I wonder if this averaging loses fine-grained alignment. Maybe you can show visualizations of the learned weights or try different aggregation methods (max or attention-weighted pooling) to see if it helps?

3. Performance gain inconsistency is not well explained. For CLIP-VG, the improvement is +7.74% on RefCOCO+ testB but only +0.71% on RefCOCO testA. Also, the gain is much smaller on SimVG, which uses stronger pretrained features. Does ToB have diminishing returns when the base model already encodes spatial context well?

4. Even though the paper claims they are plug-and-play, however, it is not clear and shown how to involve this method to MLLMs for visual grounding. Considering that MLLMs dominate in all the VL tasks, e.g. InternVL series, QwenVL series, Seed1.5VL, etc, it is necessary to clarify the motivation on building this plug-and-play module only compatible to specialist models.

**Questions:**

Please refer to my questions raised in each weakness.

---

> ### Author Response · Authors · 2025-11-23
> **Rebuttal for Reviewer Fusm**
>
> Dear Reviewer Fusm, thank you for your constructive comments. We address each weakness (W) and question (Q) below.
> ***
> **Q1:** The paper misses a lot of recent works on Visual Grounding. Therefore, the related work section is seriously incomprehensive. Tables should cite and compare SegVG (ECCV'24), AttBalance (ACM-MM), ExpVG, etc. Those are all visual grounding methods without comparison and discussion in this paper.
>
> **R1:** Thank you for pointing out the missing references. As suggested, we have added the following contents to include the discussion of three recently proposed important VG methods, namely SegVG [1], AttBalance [2], and ExpVG [3], in the **related work section**:
>
> “SegVG (Kang et al., 2024) transfers the bounding box annotation into additional segmentation signals to exploit the pixel-level details of the target regions.”
>
> “AttBalance (Kang et al., 2025a) dynamically imposes and balances constraints of the attention to optimize the behavior of visual features within language-relevant regions.”
>
> “Recent ExpVG (Kang et al., 2025b) systematically explores the paradigm and data designs for VG in multi-modal large language models (e.g., LLaVA series).”
>
> In addition, we have also presented their performance in our revised **Table 1** for direct comparison with our proposed method.
>
> [1] Kang, W., Liu, G., Shah, M., & Yan, Y. (2024, September). Segvg: Transferring object bounding box to segmentation for visual grounding. In European Conference on Computer Vision (pp. 57-75). Cham: Springer Nature Switzerland.
>
> [2] Kang, W., Zhou, L., Wu, J., Sun, C., & Yan, Y. (2025, October). Visual grounding with attention-driven constraint balancing. In Proceedings of the 33rd ACM International Conference on Multimedia (pp. 1637-1645).
>
> [3] Kang, W., Zhuang, W., Li, Z., Yan, Y., & Lyu, L. (2025). ExpVG: Investigating the Design Space of Visual Grounding in Multimodal Large Language Model. arXiv preprint arXiv:2508.08066.
> ***
> **Q2:** Textual relevance computation seems oversimplified. In Eq. (3), you just take the average correlation between image and language tokens. I wonder if this averaging loses fine-grained alignment. Maybe you can show visualizations of the learned weights or try different aggregation methods (max or attention-weighted pooling) to see if it helps?
>
> **R2:** As suggested, in **Figure 5** of the revised manuscript, we have visualized the text-aware weight $W_i$ obtained at layers 7, 8, 9, 10 in the Baseline+ToB model. As can be seen, most foreground visual tokens receive noticeably higher text-aware weights than the background ones. And moreover, for the same elephant image in the last two rows, the weight maps produced by different referring expressions emphasize different target instances, demonstrating that the ToB-generated weights can adapt appropriately to the semantics of the input text and guide the model toward the correct object. This may be attributed to the two learnable linear projectors $P_{v,i}$ and $P_{l,i}$ in Eq. (3), which provide an appropriate common space for generating more accurate visual-linguistic correlations.
>
> In addition, we have also conducted an ablation study in **Table 8** of our revised manuscript to compare different textual aggregation strategies, including average pooling, max-pooling, and attention-weighted pooling. The results suggest that the average pooling operation adopted in our ToB module achieves the best performance (80.39% on val-u and 80.36% on test-u), indicating that aggregating correlations across all linguistic tokens provides a stable and reliable signal for distinguishing foreground-relevant tokens from background ones. Moreover, it can also be found that the overall differences between different aggregation strategies are relatively small, indicating that ToB is insensitive to diverse choices of aggregation methods and different pooling operations can be flexibly adopted based on practical requirements.

---

> ### Author Response · Authors · 2025-11-23
> **Rebuttal for Reviewer Fusm**
>
> **Q3:** Performance gain inconsistency is not well explained. For CLIP-VG, the improvement is +7.74% on RefCOCO+ testB but only +0.71% on RefCOCO testA. Also, the gain is much smaller on SimVG, which uses stronger pretrained features. Does ToB have diminishing returns when the base model already encodes spatial context well?
>
> **R3:** For CLIP-VG, we agree that the performance trends of our method exhibit variations across different datasets. We believe it is primarily because **CLIP-based encoders still have substantial room for improvement on the RefCOCO+ dataset**, where the usage of location-related words (e.g., “left” or “right”) is strictly disallowed in the expression. This may pose a great challenge to CLIP encoders to achieve accurate spatial location reasoning on that dataset. This analysis can be corroborated by the observation that CLIP-VG achieves competitive results on the RefCOCO dataset compared to D-MDETR and RefFormer, but the performance gap between these methods becomes quite pronounced on the RefCOCO+ dataset, thereby affording ToB greater room for improvement.
>
> As for SimVG, we believe its great performance stems primarily from its extra pretraining process on 28K additional VG samples, so that its encoders have already adapted to the VG task. To show this, in **Table 1**, we report the results by integrating our ToB module into the original SimVG model without extra VG pretraining. In this case, although with the same encoder architecture, ToB brings much larger performance improvements. These results demonstrate that if the backbone model already possesses the capability to capture visual-textual associative information specific to VG tasks, the potential scope for ToB enhancement will naturally narrow, thereby leading to diminished gains. Nevertheless, ToB can still improve the performance of SimVG (28K), showing its effectiveness.
> ***
> **Q4:** Even though the paper claims they are plug-and-play, however, it is not clear and shown how to involve this method to MLLMs for visual grounding. Considering that MLLMs dominate in all the VL tasks, e.g. InternVL series, QwenVL series, Seed1.5VL, etc, it is necessary to clarify the motivation on building this plug-and-play module only compatible to specialist models.
>
> **R4:** We appreciate this insightful comment. Despite their outstanding performance in advanced visual-language reasoning tasks, the training and deployment costs of MLLMs remain prohibitively high. Their memory requirements during inference also exhibit a steeply increasing trend, and the parameter scale of such models potentially imposes an unbearable burden on practical resource-constrained or real-time applications. In contrast, specialist VG methods are still widely used and studied due to their efficiency, lower resource demands, and suitability for efficient transfer learning and retraining. Our plug-and-play design specifically targets these models to offer a **lightweight, resource-friendly improvement module**. Note that, as shown in **Table 1**, by integrating ToB modules into the SimVG (28K) model, the resulting SimVG(28k)+ToB model achieves comparable results to various large vision-language models. Specifically, it consistently outperforms Ferret-7B, Groma-7B, and Qwen-VL-7B across all datasets, and is competitive with Qwen2.5-VL-7B on most cases.
>
> Nevertheless, we agree that ToB is not directly compatible with current MLLM architectures. This also outlines a future direction: exploring a **text-guided multi-modal token merging strategy tailored to MLLMs** for VG tasks. We view this as a promising extension, especially as recent work such as LLaVA-PruMerge [1] has begun exploring adaptive token reduction for efficient MLLM.
>
> [1] Shang, Y., Cai, M., Xu, B., Lee, Y. J., & Yan, Y. (2025). Llava-prumerge: Adaptive token reduction for efficient large multimodal models. In Proceedings of the IEEE/CVF International Conference on Computer Vision (pp. 22857-22867).

---

> ### Author Response · Authors · 2025-11-24
>
> Dear Reviewer Fusm,
>
> Thanks for your reviews. We've tried our best to provide our responses and revise our paper.
> Could you please let us know if you need any further clarifications or discussions?
>
> We would really appreciate it if we could get your feedback.
>
> Best regards,
>
> Authors.

---

> ### Comment · Reviewer_Fusm · 2025-11-26
>
> Thanks for the rebuttal and the additional experiments.
>
> However, my main concern remains unresolved. As admitted in R4, the proposed method is not compatible with current MLLM architectures. Given that the field has largely shifted towards MLLMs (e.g., Qwen-VL, InternVL), a method restricted to specialist models has limited contribution and impact in the current landscape.
>
> Therefore, I will maintain my original rating.

---

> ### Author Response · Authors · 2025-11-28
> **Response to Reviewer Fusm**
>
> **Comments:** As admitted in R4, the proposed method is not compatible with current MLLM architectures. Given that the field has largely shifted towards MLLMs (e.g., Qwen-VL, InternVL), a method restricted to specialist models has limited contribution and impact in the current landscape.
>
> **R:** We cannot accept the reviewer's comments that our work makes only a limited contribution **simply because “it is not compatible with current MLLMs”**. The reasons are as follows:
>
> 1. We have clearly stated the main contribution of the proposed ToB module, which is to **develop the first text-guided multi-modal token merging strategy** that can fully exploit the textual relevance information during the merging decision process. The other two reviewers also gave positive comments on this design:
>
>      **Reviewer bSxz:** “Unlike prior token merging approaches that rely solely on visual similarity, ToB incorporates textual relevance into the merging decision. This dual-modality criterion is conceptually novel and well-motivated for visual grounding tasks.”
>
>      **Reviewer weHD:** “First to jointly use visual similarity and textual relevance for token merging in VG.”
>
> 2. The reason why our ToB module is incompatible with **CURRENT** MLLMs (e.g., Qwen-VL, InternVL) is mainly because that they all utilize the GPT-like structures. We cannot efficiently extract text-only information from these models to calculate the text relevance in our ToB module. However, the proposed ToB modules can be easily incorporated into **ANY two-tower architecture with separate textual and visual encoders** in a plug-and-play manner, **no matter whether they are MLLMs or specialist models.** This compatibility has already been demonstrated by our experiments on three different benchmark models. Science and technology develop in a spiral pattern. If we have a more powerful MLLM model employing the two-tower architecture in the future, our ToB module will be compatible with that.
>
> 3. Moreover, as shown in Table 1, by integrating ToB modules into the SimVG (28K) model, the resulting SimVG(28k)+ToB model achieves comparable results to various MLLMs. Specifically, **it consistently outperforms Ferret-7B, Groma-7B, and Qwen-VL-7B across all datasets, and is competitive with Qwen2.5-VL-7B on most cases.** This also indicates the effectiveness and practicality of specialist models, especially considering their development efficiency.
>
> 4. **MLLMs are not everything**. If an approach is arbitrarily deemed to offer limited contribution solely because “it is incompatible with current MLLMs”, we ought to stop all scientific exploration unrelated to MLLMs, including research into new architectures for MLLMs themselves.

---

### Official Review · Reviewer_bSxz · 2025-10-31

**Soundness:** 3
**Presentation:** 3
**Contribution:** 2
**Rating:** 4
**Confidence:** 4

**Summary:**

This paper presents Token Blurring (ToB), a plug-and-play module designed to improve visual grounding by dynamically merging image tokens that are visually redundant and text-irrelevant. ToB first computes pair-wise visual similarity and text-based relevance for image tokens, then selects the top-r background-like token pairs to merge via a bipartite matching scheme. he method is simple, model-agnostic, and effective in improving grounding performance, though its claimed efficiency benefits are not empirically validated in the paper.

**Strengths:**

(1)ToB can be seamlessly inserted into existing visual grounding pipelines without modifying model architectures or requiring additional supervision, making it practical and widely applicable.

(2)Unlike prior token merging approaches that rely solely on visual similarity, ToB incorporates textual relevance into the merging decision. This dual-modality criterion is conceptually novel and well-motivated for visual grounding tasks.

(3)ToB demonstrates consistent performance gains when applied to weak (CLIP-VG), medium-sized (DINOv2-based baseline), and strong models (SimVG with BEiT-3). This cross-model effectiveness indicates strong generalizability.

**Weaknesses:**

(1)The authors highlight efficiency improvement as a key advantage of ToB, but the paper does not provide any quantitative experiments (e.g., inference speed, FLOPs reduction, or memory savings) to support this claim. Since the merging process itself adds computation (pair-wise similarity, textual relevance, ranking), it is unclear whether the overall pipeline is actually faster.

(2)The method assumes that background tokens are visually similar and text-irrelevant, which does not hold in many realistic visual grounding scenarios involving multiple similar objects or cluttered scenes.

(3)W_i is computed through AvgPool of text–vision correlations, yielding only a scalar per token. This severely limits the ability to distinguish foreground from background, making the merging decision potentially unreliable.

(4)The A/B splitting mechanism is heuristic and lacks theoretical or empirical justification. This operation may disrupt spatial locality and lead to unstable token pairing.

(5)Token averaging destroys spatial and boundary information critical for VG tasks, but the paper does not analyze its effect on bounding box regression or small-object localization.

(6)The merging policy is rule-based rather than learned, raising concerns about adaptability across diverse scenes.

**Questions:**

(1)Absence of efficiency evaluation despite claimed benefits.

(2)Potential loss of spatial and structural information.

(3)Non-learnable, heuristic merging policy.

---

> ### Author Response · Authors · 2025-11-23
> **Rebuttal for Reviewer bSxz**
>
> Dear Reviewer bSxz, thank you for your constructive comments. We address each weakness (W) and question (Q) below.
> ***
> **W1 & Q1:** The authors highlight efficiency improvement as a key advantage of ToB, but the paper does not provide any quantitative experiments (e.g., inference speed, FLOPs reduction, or memory savings) to support this claim. Since the merging process itself adds computation (pair-wise similarity, textual relevance, ranking), it is unclear whether the overall pipeline is actually faster.
>
> **R1:** As suggested, we have added a detailed efficiency analysis in **Table 5** of our revised version. As can be seen, although the merging process introduces additional operations, our ToB modules consistently enhance the overall efficiency of the Baseline model across different implementation settings, in terms of computational overhead (GFLOPs), memory usage (GB), and inference speed (fps). **Such improvements become even more significant when the number of blurring tokens $r$ in each ToB module increases from 64 to 196**. While the grounding accuracy of Baseline+ToB drops as $r$ increases, the worst-performed variant achieved at $r=196$ still outperforms the Baseline model by 2.24% (78.91% vs. 76.67%) and 2.72% (78.77% vs. 76.05%) on the two splits, respectively, and also surpasses TransVG and MaPPER employing the same encoders (see Table 1). This demonstrates that integrating our ToB modules can bring benefits to the efficiency and effectiveness of VG models.
> ***
> **W2:** The method assumes that background tokens are visually similar and text-irrelevant, which does not hold in many realistic visual grounding scenarios involving multiple similar objects or cluttered scenes.
>
> **R2:** It is worth noting that the main purpose of our ToB module is to reduce the negative influence of **text-irrelevant background tokens** instead of finding and removing all background tokens. As shown in Figure 6 of our revised version, in the scenarios involving multiple background objects that are visually similar to the target object, ToB tends to preserve portions of information for all these objects since they may also possess significant textual relevance. But in contrast, those background tokens that are less relevant to the referring expression can still be successfully merged. This phenomenon is more obvious when $r=196$. The compression of these text-irrelevant tokens will reduce the number of negative-impact tokens involved in the attention mechanism, which allows the model to focus more on distinguishing the target object from those visually similar background objects, thus leading to better grounding performance.
> ***
> **W3:** W_i is computed through AvgPool of text–vision correlations, yielding only a scalar per token. This severely limits the ability to distinguish foreground from background, making the merging decision potentially unreliable.
>
> **R3:** In **Figure 5** of our revised version, we have visualized the text-aware weight $W_i$ obtained at layers 7, 8, 9, 10 in the Baseline+ToB model. As can be seen, most foreground visual tokens receive noticeably higher text-aware weights than those background ones. This indicates that our average pooling operation **DOES NOT** limit the ability of our ToB modules in distinguishing foreground from background. This may be attributed to the two learnable linear projectors $P_{v,i}$ and $P_{l,i}$ in Eq. (3), which provide an appropriate common space for generating more accurate visual-linguistic correlations. Moreover, for the same elephant image in the last two rows, the weight maps produced by different referring expressions emphasize different target instances, demonstrating that the ToB-generated weights can adapt appropriately to the semantics of the input text and guide the model toward the correct object.

---

> ### Author Response · Authors · 2025-11-23
> **Rebuttal for Reviewer bSxz**
>
> **W4:** W_i is computed through AvgPool of text–vision correlations, yielding only a scalar per token. This severely limits the ability to distinguish foreground from background, making the merging decision potentially unreliable.
>
> **R4:** In our opinion, compared with the text-relevance, we believe that the spatial locality is less important for solving VG tasks. The text-relevance criterion encourages the text-irrelevant tokens to be merged together. These tokens can be more likely to be background tokens, since they are not aligned with the input referring expression. In this case, even though the merged tokens are not located in a contiguous local area (i.e., with less spatial locality), the number of text-irrelevant tokens involved in the attention calculation is still actually reduced. This is consistent with the objectives of our proposed ToB module, whose main contribution is to design a novel multi-modal token merging strategy to reduce the negative influence of some background tokens. Therefore, the A/B splitting mechanism is effective enough for our method. Nevertheless, as suggested, we may explore the role of spatial locality in addressing VG tasks in our future work.
> ***
> **W5 & Q2:** Token averaging destroys spatial and boundary information critical for VG tasks, but the paper does not analyze its effect on bounding box regression or small-object localization.
>
> **R5:** To investigate the effect of our ToB module on precise bounding box regression and small-object localization, we conduct a fine-grained evaluation in **Table 6** of our revised version. We present the grounding accuracy achieved under stricter IoU thresholds and across different object scales. As can be seen, compared to the Baseline model, ToB consistently brings improvements across all IoU thresholds. When $r = 64$, the performance on the val-u split increases from 76.67% to 80.39% at 0.5 IoU, and such gains become even more pronounced at higher thresholds (e.g., from 54.34% to 65.22% at 0.8 IoU and from 24.83% to 47.49% at 0.9 IoU). Similar trends can also be observed on the test-u split and under the $r=196$ setting. This suggests that the bounding boxes predicted by our Baseline+ToB model are more consistent with the ground-truth boxes, and demonstrates that suppressing text-irrelevant background tokens helps the model maintain more accurate visual-textual alignment.
>
> Moreover, we divide the target objects into three groups, namely small ($<128\times128$ pixels), medium ($128\times128$–$256\times256$ pixels), and large ($>256\times256$ pixels) objects, based on their bounding-box area. The results in Table 6 indicated that ToB yields substantial improvements, particularly on localizing small objects where background clutter tends to dominate the overall token distribution.
>
> All the results mentioned above confirm that our ToB module can preserve the fine-grained spatial information required for high-precision bounding box regression and small-object localization.
> ***
> **W6 & Q3:** The merging policy is rule-based rather than learned, raising concerns about adaptability across diverse scenes.
>
> **R6:** Actually, each ToB module contains two projectors $P_{v,i}$ and $P_{l,i}$ that require to be learned (see Eq. (3)). These two projectors play a key role in constructing a common space to extract correct visual-linguistic correlations for generating the text-aware weights. These weights are not only used to calculate the text-relevance of each token pair, but also utilized to emphasize the text-referred areas in visual features (see Eq. (8)). The gradients of detection loss propagate backwards through $W_i'$ in Eq. (8) to $P_{v,i}$ and $P_{l,i}$ will make these two projectors more sensitive to the visual-textual alignment. Therefore, our ToB module is able to benefit from the fine-tuning process, allowing it to adapt to diverse scenarios. In addition, the results across three datasets in Table 1 validate the adaptability and effectiveness of ToB.

---

> ### Author Response · Authors · 2025-11-24
>
> Dear Reviewer bSxz,
>
> Thanks for your reviews. We've tried our best to provide our responses and revise our paper.
> Could you please let us know if you need any further clarifications or discussions?
>
> We would really appreciate it if we could get your feedback.
>
> Best regards,
>
> Authors.

---

> > ### Comment · Reviewer_bSxz · 2025-11-26
> >
> > Thank you for the response and new experiments. I still have the following concerns:
> >
> > (1) In Table 5, why do the authors not directly compare the performance of Baseline+ToB and ToMe+ToB under the same r value? Instead, why do they only report the result for ToMe+ToB at r = 96?  As claimed by the authors, “When the blurring token number r in each ToB module increases from 64 to 196, resulting in a more significant efficiency improvement over the Baseline model.” For RefCOCOg, maybe Baseline+ToB is more efficient than ToMe+ToB with r = 96 (just a guess).
> >
> > (2) In the second row of Figure 5, the visualizations of layer 10 are weird.  As the layers deepen, the weight assigned to the target object is expected to become increasingly prominent compared to the background. However, Row 2 appears to show the opposite trend.

---

> ### Author Response · Authors · 2025-11-28
> **Response to the concerns**
>
> **Q1:** In Table 5, why do the authors not directly compare the performance of Baseline+ToB and ToMe+ToB under the same r value? Instead, why do they only report the result for ToMe+ToB at r = 96? As claimed by the authors, “When the blurring token number r in each ToB module increases from 64 to 196, resulting in a more significant efficiency improvement over the Baseline model.” For RefCOCOg, maybe Baseline+ToB is more efficient than ToMe+ToB with r = 96 (just a guess).
>
> **R1:** For ToMe+ToB, we additionally insert ToMe modules into the first 6 visual encoder layers of Baseline+ToB. Therefore, the token merging operations are performed in all 12 layers of ToMe+ToB. In this way, **if we set the same value to $r$ for both models, there will be many fewer tokens remaining in the last 6 layers of ToMe+ToB than those of Baseline+ToB, resulting in an unfair efficiency comparison**. Therefore, we set $r=96$ for ToMe+ToB to achieve a balance between its performance and efficiency. From Table 5, we can see that ToMe+ToB performs slightly worse than Baseline+ToB ($r=196$) but can be more efficient. This indicates that our ToB module is compatible with other token compression strategies, allowing users to select appropriate configurations based on their application scenarios (e.g., prioritizing efficiency or performance).
> ***
> **Q2:** In the second row of Figure 5, the visualizations of layer 10 are weird. As the layers deepen, the weight assigned to the target object is expected to become increasingly prominent compared to the background. However, Row 2 appears to show the opposite trend.
>
> **R2:** We analyze the reason for the results in the second row of Figure 5 as follows: From layers 7 to 10, the model is able to focus more on the image areas associated with the expression of “**wearing a blue shirt**”, but incorrectly identifies those relevant to “**the woman**” (e.g., arm and hand regions) as background. This may be because the word embedding for “the woman” generated by BERT-B (i.e., the linguistic encoder used in our Baseline+ToB model) fails to align well with the human body visual tokens **lacking gender-specific features** (e.g., without facial information). Nevertheless, other visualization results still demonstrate that our ToB method can distinguish text-irrelevant tokens from foreground ones, thus effectively mitigating their negative effects on target object detection.

---

### Author Response · Authors · 2025-11-21
**Summary of Revisions**

We thank the reviewers for their constructive feedback. In this rebuttal revision, we have substantially improved the paper by incorporating new analyses, experiments, and visualizations, all marked in **blue** in the manuscript. The main updates are as follows:

1. **Detailed efficiency analysis (Table 5).**
   We added a detailed discussion of the computational efficiency of ToB, including FLOPs, memory usage, latency, and cumulative token reduction across layers, along with comparison to the hybrid ToMe+ToB setting.

2. **Fine-grained evaluation results (Table 6).**
   We provided evaluation results under stricter IoU thresholds (0.5–0.9) and across object scales (small, medium, large), offering a more comprehensive understanding of ToB’s effect on high-precision bounding box regression and small-object localization.

3. **Ablation on aggregation methods (Table 8).**
   We compared different aggregation methods for textual relevance, including average pooling, max pooling, and attention-weighted pooling, to demonstrate the robustness of ToB to the choice of textual aggregation.

4. **Additional qualitative visualizations (Figures 5 and 6).**
   We added extensive new visualizations, including (i) textual relevance maps that highlight visual–textual alignment, and (ii) token merging results in scenes with multiple visually similar objects, showing how ToB selectively preserves expression-relevant regions.

5. **Updated comparison with state-of-the-art methods (Revised Table 1).**
   We added several important recent works (e.g., SegVG, AttBalance, ExpVG, etc.) to improve completeness. We also supplemented the experimental results on SimVG without extra visual grounding pretraining to demonstrate the effectiveness of our method.

We apologize for the delayed response. The revision took longer than expected as we conducted additional experiments and analyses to address the reviewers’ comments thoroughly. A detailed, point-by-point response to each reviewer will follow.

---

### Comment · Area_Chair_jNcb · 2025-11-26
**A Reminder on Your Crucial Role in the ICLR Discussion Period**

Dear Reviewers:

As the Area Chair, I would like to sincerely thank you for the time and expertise you have invested in writing your initial review. Your insights are invaluable to the decision-making process.

We are now entering the critical discussion and rebuttal phase. This is a collaborative process where authors have the opportunity to address your concerns and questions. Your active participation in this phase is essential to ensure we reach a fair and well-informed final decision.

I strongly encourage you to:

Engage with the Authors' Rebuttal: Please read the authors' response carefully and substantively.

Participate in the Discussion: Engage with the other reviewers on the forum. If the authors have clarified a point, please acknowledge it. If you have follow-up questions or remaining concerns, please voice them. Your dialogue with fellow reviewers is key to reaching a consensus.

Update Your Review (if necessary): Based on the discussion and rebuttal, you may feel the need to adjust your score or final recommendation. Please do so, as it reflects a more holistic view of the paper.

Your continued engagement ensures the integrity and quality of the ICLR conference. Thank you for your vital contribution to our community.

Best regards,

Area Chair, ICLR 2026

---

### Meta-Review · Area_Chair_2Rre · 2026-01-04

**Summary:**

The paper received two scores marginally below the acceptance threshold and one marginally above it. Reviewers raised multiple concerns regarding the experimental results and the method’s compatibility with current MLLM architectures. After reviewing the authors’ responses, many of the issues raised by the reviewers remain unclear. The paper requires significant improvement before it can be considered for acceptance.

**Reviewer Concerns:**

In my opinion, concerns regarding the experimental results and the method’s compatibility with current MLLM architectures have not been adequately addressed.

**Reviewer Scores:**

Based on the rebuttal, it is unlikely that the reviewers will revise their scores to positive ratings.

---

### Decision · Program_Chairs · 2026-01-26

Reject